# Study of the Early Effects of Chitosan Nanoparticles with Glutathione in Rats with Osteoarthrosis

**DOI:** 10.3390/pharmaceutics15082172

**Published:** 2023-08-21

**Authors:** Patricia Ramírez-Noguera, Iliane Zetina Marín, Blanca Margarita Gómez Chavarin, Moisés Eduardo Valderrama, Laura Denise López-Barrera, Roberto Díaz-Torres

**Affiliations:** 1Multidisciplinary Research Unit, Facultad de Estudios Superiores Cuautitlán, Universidad Nacional Autónoma de México, Carretera Cuautitlán-Teoloyucan Km. 2.5, San Sebastián Xhala, Cuautitlán Izcalli CP 54714, Mexico; ramireznoguera@unam.mx (P.R.-N.);; 2School of Medicine, Universidad Nacional Autónoma de México, Circuito Interior, Ciudad Universitaria, Av. Universidad 3000, Mexico City CP 04510, Mexico; 3Equine Hospital, Facultad de Estudios Superiores Cuautitlán, Universidad Nacional Autónoma de México, Carretera Cuautitlán-Teoloyucan Km. 2.5, San Sebastián Xhala, Cuautitlán Izcalli CP 54714, Mexico

**Keywords:** osteoarthrosis, glutathione nanoparticles, chitosan, chondrocytes, oxidative stress

## Abstract

Due to cartilage’s limited capacity for regeneration, numerous studies have been conducted to find new drugs that modify osteoarthrosis’s progression. Some evidence showed the capability of chitosan nanoparticles with glutathione (Np-GSH) to regulate the oxide-redox status in vitro in human chondrocytes. This work aimed to evaluate the capacity of Np-GSH in vivo, using Wistar rats with induced surgical osteoarthritis. Radiographic, biochemical (GSH and TBARS quantification), histopathological, and immunohistochemical (Col-2 and MMP-13) analyses were performed to evaluate the progress of the osteoarthritic lesions after the administration of a single dose of Np-GSH. According to the results obtained, the GSH contained in the NPs could be vectored to chondrocytes and used by the cell to modulate the oxidative state reduction, decreasing the production of ROS and free radicals induced by agents oxidizing xenobiotics, increasing GSH levels, as well as the activity of GPx, and decreasing lipid peroxidation. These results are significant since the synthesis of GSH develops exclusively in the cell cytoplasm, and its quantity under an oxidation–reduction imbalance may be defective. Therefore, the results allow us to consider these nanostructures as a helpful study tool to reduce the damage associated with oxidative stress in various diseases such as osteoarthritis.

## 1. Introduction

Osteoarthritis is a degenerative, progressive, and disabling articular cartilage disease accompanied by underlying bone and soft tissue changes. It is mainly characterized by the progressive erosion of the articular cartilage, where the same inflammatory process causes the degradation of the extracellular matrix and the death of the chondrocytes.

Cartilage is characterized by having a minimal capacity for repair. The inflammatory process favors the deterioration of the joint due to the release of metalloproteinases, inflammatory mediators, and cytokines, which degrade proteoglycans and decrease their synthesis, causing depletion of the cartilage matrix [1,2].

On the other hand, the ischemia and reperfusion generated by arthritis lead to the production of free radicals and reactive oxygen species, which degrade the hyaluronic acid and glycosaminoglycan chains, inhibiting their synthesis, increasing the production of cytokines and metalloproteinases, and inducing apoptosis of chondrocytes [3].

Currently, there is no effective treatment to stop the progression of this disease [4]. Joint lesions are conventionally treated with non-steroidal anti-inflammatory drugs (NSAIDs) and glycosaminoglycans to prevent the inflammatory process, reduce pain, and improve joint function. However, they only represent a symptomatic treatment, the effectiveness of which is still debated [5,6,7,8].

Recent studies have suggested using antioxidant compounds within the synovial fluid to treat osteoarthritis, specifically to modulate the oxidative process rather than eliminate free radicals or suppress the inflammatory process [3,7,9,10].

In this research, we studied the effect of intra-articular administration of chitosan nanoparticles with glutathione in rats with surgically induced osteoarthritis through dissection of the cruciate ligament in the knee of male Wistar rats. This method promotes highly reproducible lesions with little variability and rapid progress. It homogenizes the wide range of existing variables, making it ideal for short-term studies. It allows observing subtle effects of the drug that might otherwise go unnoticed in the face of accelerated degeneration [11,12]. Glutathione (GSH) is the primary cellular antioxidant. Its synthesis occurs exclusively in the cell cytoplasm, modulating oxidative stress in chondrocytes and their resistance and resilience to disease. The existence of nanostructured systems capable of promoting antioxidant events that improve the histopathological, cellular, and biochemical condition associated with cartilage and with the extracellular matrix in osteoarthritis represents an essential possibility of use for patients with this disease.

It is considered that the cellular events that contribute to osteoarthritis are the oxidation–reduction imbalance, inflammation, and the low contribution of antioxidant complexes to controlling this state, as well as the disorganization and function of proteoglycans and collagen of the ECM, where the contribution of antioxidant activity is essential.

## 2. Materials and Methods

### 2.1. Nanoparticles

Chitosan nanoparticles with glutathione (NP-GSH) were prepared by the ionic gelation method [13], from a 1% acetic acid-based solution adjusted to a pH of 4.1 and with 1% Pluronic F-68. The solution was divided into 2, and in the first one, 0.3% chitosan and 1% GSH were solubilized; the second solution included 0.1% sodium tripolyphosphate (STPP). Before the solutions were mixed, both were independently filtered through paper to remove insoluble impurities. Both solutions were mixed in a 1:1 ratio, adding the STPP solution to the chitosan solution and keeping them agitated for 4 h. To acquire a concentration of chitosan comparable to hyaluronic acid (10 mg/mL), used as control treatment in this study, the already-formed nanoparticles were ultracentrifuged at 22,253 g/4 °C/60 min. After the above, the formed ring was recovered and washed 4 times with 1% acetic acid solution, adjusting the pH to 4.1 to eliminate excess materials that remained without forming nanoparticles.

Unlike the methods in previous studies, in this study, the systems used were filtered through a membrane. To sterilize the nanoparticles in this study, before their administration, they were exposed to ultraviolet light for 14 h and kept refrigerated (4 °C) until use. The chitosan nanoparticles (NP-Q) were prepared using the same methodology, omitting only the addition of GSH.

For nanoparticle characterization, particle size and zeta potential were determined using a Zetasizer and a Nanosight NS300 instrument from Malvern Panalytical. Transmission electron microscopy (TEM) was performed using a tungsten filament HV at 100 kV (Hitachi, Tokyo, Japan). Quantification of GSH inside nanoparticles was conducted indirectly via a colorimetric method using 2,2′-dinitro-5-5′dithiodibenzoic acid (DTNB), and the results were determined at a wavelength of 425 nm.

### 2.2. Induction of Osteoarthritis

All the studies were carried out following the protocol approved by CICUAE (FESC/CICUAE/06/03/2018) for managing experimental animals from FES-Cuautitlán, UNAM. Fifty male Wistar strain Rattus norvegicus rats (280–320 g) were obtained from the Isolation and Vivarium Unit of the FESC, Multidisciplinary Research Unit. Male rats weighing 280–320 g were placed in boxes with 5 individuals. They had food and water ad libitum in the UIM Vivarium, respecting the environmental light–dark cycles. They were randomly separated into 4 groups (Table 1) according to the treatment and subjected to 2 -h monitoring. In pre-surgical preparation, the animals were transferred to the operating room, where they were administered ketamine–xylazine (50 mg–10 mg/kg/IM) for general anesthesia. Subsequently, gentamicin (6 mg/kg/IM) and buprenorphine (0.05 mg/kg/SC) were administered. Each animal was weighed, and X-rays were taken before the implementation of the protocol of trichotomy and embroidering of the knee to be operated on. The knee to be used was randomly chosen for each rat.

Any animal showing pain or lameness unrelated to the induced lesion, infection, or other pathology was eliminated from the study. Anterior cruciate ligament transection (ACLT) was performed as described by Hayami and Pickarski (2004) [14] by medial incision, followed by lateral dislocation of the patella, and placing the joint in flexion to expose and transect the cranial cruciate ligament. Unlike meniscectomy, the osteoarthritis lesion was induced by ACLT with limited or partial cartilage destruction. 

This procedure promotes an experimental model with excellent reproducibility and is helpful in short-term disease studies [15]. After 7 days after the surgical intervention and the corresponding treatment, the animals were brought to the endpoint, and both knees were collected for further studies. 

The experimental groups comprised 10 rats with the confirmed lesion. They were administered intra-articular treatments, with a single dose of each treatment, 7 days after surgery, using 40 µL of each treatment and an insulin syringe. The treatments used were as follows: negative control 1% acetic acid, Hartmann solution (HS), 10 mg/mL hyaluronic acid (HA), NP-GSH, and NP-Q.

#### 2.2.1. Radiographic Study

Computed radiology studies (Computed Radiography FCR Prima (Fujifilm, Tokyo, Japan)) were performed before surgery, treatment administration, and sacrifice, taking two shots per rat (mediolateral and craniocaudal) for both knees to assess the degree of osteoarthritis lesion.

#### 2.2.2. Sacrifice and Sample Processing

At 14 days post-surgery, the animals were sacrificed under anesthesia following the same protocol as for surgery and brought to the endpoint to recover both knees. The side contralateral to the lesion was considered the control, and the ACLT was the experimental side. Five rats from each group were destined for the histopathological study, maintaining the integrity of the capsule. At the same time, the articular cartilage was obtained from the remaining rats for the biochemical analysis of GSH and TBARS quantification. According to the Bradford method, a portion of the sample was used to determine the total protein concentration. The entire content of GSH was quantified by spectrophotometry [16].

### 2.3. Lipoperoxidation

The lipid peroxidation assay was carried out according to the method described by Ohkawa et al. (1979) with some modifications [17].

### 2.4. Histopathological and Immunohistochemical Evaluation

Considering joints as complex structures of different tissues composed of a bone base covered on their articular surface by a specialized tissue and articular cartilage, which is composed mostly of water (60–80%) and an extracellular matrix formed by type II collagen fibers (>60%), sunk in a gel of a fundamental substance rich in proteoglycans (PGs) and glycoproteins (GPs), synthesized by the only resident cell of the cartilage, the chondrocyte immersed in the extracellular matrix (cartilaginous matrix), the histopathological analysis of the joint exposed to the Np under study allows us to know the effects of exposure in an integral and specific way for some constituents.

As primary staining for the histopathological examination, hematoxylin–eosin staining was performed, along with van Gieson, toluidine blue, Safranin O, and acridine orange staining. Then, immunohistochemical tests were also performed for collagen type 2A1 (Col-2) and metalloproteinase 13 (MMP-13).

The histopathological sections were evaluated using a guide modified from the Mankin system qualifying each morphological characteristic individually [18,19,20].

#### Immunohistochemistry

Tissue samples were evaluated for indirect immunostaining with the mouse monoclonal antibodies Col2A1 (M2139) and MMP-13 (C-3) from Santa Cruz Biotechnology, using a biotinylated anti-mouse secondary antibody (Donkey anti-mouse IgG (H + L) Biotin, Millipore, Burlington, MA, USA) and the avidin–biotin complex (ABC Elite PK-6100, Vector, Burlingame, CA, USA). Antibody binding was revealed with the DAB-peroxidase kit (Peroxidase Substrate Kit DAB SK-4100 Vector) and counterstaining with Crystal Violet (C5042, Sigma, St. Louis, MO, USA).

The presence of type II collagen (Col2A1) is related to the presence of healthy tissue, which, when hypertrophied after damage, begins to be replaced by type X collagen. On the other hand, metalloproteinase 13 (MMP- 13) is the most widely used marker of inflammation in the evaluation of cartilage, as it is the main protease in articular cartilage degradation [21,22].

### 2.5. Statistical Analysis

For biochemical tests, each experiment was performed in triplicate. Data are expressed as mean ± SEM and were analyzed using a one-way analysis of variance (ANOVA) followed by a post hoc test of Tukey’s multiple comparisons of means with the Prism program ver 10.0.2 (Graphpath), considering significant differences if *p* < 0.05. In the case of the radiographic and histopathological analysis of each characteristic, the data were analyzed using a two-way analysis of variance (ANOVA), followed by a post hoc test of multiple comparisons of Tukey means, considering significant differences if *p* < 0.05. In addition to the above, the characteristics of the histopathological examination were all analyzed together, using a one-way analysis of variance (ANOVA) followed by a post hoc test of multiple comparisons of Kruskal–Wallis means, considering significant differences if *p* < 0.05.

## 3. Results and Discussion

### 3.1. Preparation and Characterization of Nanoparticles

Two sets of chitosan NPs, with and without glutathione, were prepared using the ion gelation method previously described [13]. The results of their characterization are shown in Figure 1 and Figure 2 and Table 2

### 3.2. Estimation of the Concentration of Chitosan and GSH

Once the nanoparticles were prepared and conditioned for their administration to the rats with osteoarthritis, we quantified the concentrations of the components to perform the intra-articular administration based on the body weight of the experimental rats. The chitosan concentration in the nanoparticles was 4.436 mg/mL, and GSH 11.3 mM tested was estimated by spectrophotometry. Then, the administered intraarticular doses were 0.6 chitosan mg/Kg in NP-Q and NP-GSH (0.46 mg/kg GSH and 1.3 HA mg/kg).

### 3.3. Assessment of Rats with Osteoarthritis

The rats showed severe claudication one day after surgery (Figure 3), progressively decreasing until day 4 post-surgery. The present claudication and inflammation resulted from the surgical procedure, which involved the incision of the joint capsule. Subsequently, in the remaining 10 days, the rats showed no apparent discomfort in supporting the injured limb.

None of the animals showed any adverse reaction to the treatments administered. The surgical induction of OA from ACLT generated a moderate lesion after 2 weeks, confirmed by radiographic evaluation (Figure 3), which was ideal for evaluating the disease’s acute changes.

#### Radiographic Evaluation

Radiographic studies of the rats (Figure 4) were performed at 0, 7, and 14 days post-surgery, corresponding to the periods before surgical induction of OA, treatment administration, and sacrifice, respectively.

On day 0, none of the animals showed any lesions, and all received a score of “0” on the Kellgren and Lawrence scale [23]. After surgical induction of OA on day 7, the rats showed radiographic changes in the knee with ACLT that consisted of irregularities on the articular surface, osteophytes, and changes in density suggestive of sclerosis. All with an average score of “2” compared to the contralateral knees are considered controls in which no differences were observed.

In the radiographic evaluation on day 14, the ACLT groups showed more severe radiographic changes, with scores of 2 to 3.5 according to the scale of Kellgren and Lawrence [23] (Figure 5). Significant differences were found in the animals treated with Np-Q, while the animals treated with HA were the ones that best preserved the bone structure between the knees with ACLT. Although there were significant differences between those treated with Np-Q and AH, they did not show significant differences compared to the other treatments with ACLT. 

The rats treated with Np-GSH showed slightly non-significant changes compared to those treated with AH and Hartmann solution (*p* ≤ 0.001). Significant radiographic changes were observed in the control knees after administration of the different treatments. Animals treated with Hartmann solution, acetic acid, and HA did not show radiographic changes (Figure 5).

These could be associated with the onset of mechanical stress due to the compensatory effort of the contralateral knee or be part of the inflammatory process due to the arthrocentesis and not due to the treatment administered.

In the analysis of the radiological changes in the groups treated with the NP, they showed significant changes (*p* ≤ 0.001) compared to the other control groups (Figure 5). These changes ranged from an irregular articular surface to the presence of images associated with the presence of osteophytes. The control group treated with Np-GSH showed changes with grades of 1 (mild damage). In contrast, the control group with Np-Q had a much greater degeneration, reaching grade 2 lesions (moderate damage), and did not show statistically significant differences compared to the ACLT groups with AH and Hartmann solution (*p* ≤ 0.001).

### 3.4. Glutathione Quantification

The concentration of GSH and TBARS within the joint was quantified from homogenates and lysed articular cartilage and menisci.

Figure 6 shows the highest concentration of GSH in the control group with acetic acid, which is the dispersion media of the nanoparticles. The marked increase in GSH in this group is attributed to the activation of cellular mechanisms against the direct damage from exposure to this medium.

These effects were not observed in the systems containing the NPs with acetic acid as a vehicle. Additionally, although the groups with joint damage showed a decrease in GSH due to the oxidative stress that occurs during the disease, they did not show significant differences compared to the control with Hartmann solution (*p* ≤ 0.0001).

Contrary to expectations, the lowest GSH concentrations were observed in joints exposed to Np-GSH, both in control and LCC rats. No significant statistical differences were found for the other groups. Similar results were observed in a study with in vitro chondrocytes [13]. High concentrations of Np-GSH (0.6 mM GSH) showed lower GSH concentrations than those in untreated cells, without the differences being statistically significant between the two groups. These effects could be related to the response of the tissue to the inflammation caused by the surgery, modifying the movement of the experimental animals, and the cellular response in the affected area.

### 3.5. TBARS Quantification

Contrary to what has been reported in other studies [13,24,25,26], we observed that the groups without OA lesions showed the highest MDA concentrations when evaluated by the spectrophotometric method for TBARS (Figure 7).

The group treated only with acetic acid showed the highest MDA concentration (3.271 nmol/mg protein), probably due to the acidic microenvironment associated with the administered vehicle; however, this effect was not observed in joints exposed to chitosan NPs (Figure 7). This suggests a protective effect of NPs due to the antioxidant effects (13). Some works show the antioxidant capacity and the acid–base modulation of chitosan, so it likely induces a lower concentration of MDA (1349 nmol/mg protein).

### 3.6. Cellular Distribution

Hematoxylin–eosin (HE) staining was performed, as a basic stain for the morphological evaluation of the cells and the cartilaginous matrix, which evaluated the following characteristics: cell distribution, population, morphology cell, subchondral bone, tide mark, articular surface, and MEC. To evaluate the presence and distribution of collagen fibers, we used van Gieson staining, based on the selectivity of the fuchsin dye for collagen. The distribution of resident chondrocytes in the articular cartilage, under the stimulus of stress, starts from a columnar organization forming small groups of cells (isogenic groups). Later, if the stress continues, cell loss by apoptosis promotes the presence of vast spaces between cells, leaving them isolated. This change in distribution is very evident between the control groups and the groups with ACLT, where the loss of columnar organization was more evident, with a differential distribution in the areas analyzed (ANOVA; disease factor: F(1,30) = 66.59; *p* < 0.0001).

As can be seen both in Figure 8 and Figure 9 (ANOVA; treatment factor: F(4,30) = 13.52; *p* < 0.0001 and interaction: F(4,30) = 1.191; *p* = 0.3348), the loss of organization of the healthy cartilage treated with Np-Q is comparable to that of the groups with ACLT. The Np-Q affected the cellular distribution in the cartilage. This effect was not observed in the animals treated with Np-GSH, which showed an ability to preserve cellular organization in OA compared to the control group of animals exposed to Hartmann solution that was very similar to the effects induced by conventional treatment with HA.

### 3.7. Cell Population

In experimental rats with ACLT, the reduction in the resident cell population of the cartilage is evident not only in the decrease in the number of cells but also in their distribution. They can be found as isolated cells in areas utterly devoid of cellularity. The changes observed in the different treatment groups are associated with those observed in cell distribution (see Figure 8). The areas with the most significant changes in the ECM are observed in the acellular regions of the cartilage.

Although no statistical differences were observed between the treatments when analyzing the cell population in the joints of the experimental animals (Figure 9) (ANOVA; disease factor: F(1,27) = 33.52; *p* < 0.0001, treatment factor F(4,27) = 1.6; *p* = 0.2029 and interaction: F(4,27) = 0.6332; *p* = 0.6432), it should be noted that in the histopathological examination of the cartilage with ACLT after the administration of Np-GSH and for AH, completely acellular areas were not observed. The cells were widely distributed. It is worth mentioning that the animals treated with Np-GSH presented a more significant number of isogenic groups, probably as a response of the cartilage to cell proliferation and to repopulate the areas of cells lost due to the loss of the ligament in surgery (see Figure 8).

### 3.8. Cell Morphology

Osteoarthritic chondrocytes tend to undergo hypertrophy, causing alterations in the production of the ECM, which subsequently contributes to its weakness and degradation (Figure 8). A more significant number of hypertrophic chondrocytes speaks of the degree of damage to the joint and the attempts to repair it.

Among the different treatments administered, only animals treated with Np-Q showed an alteration in the morphology of the chondrocytes, both in the increase in the number of hypertrophic chondrocytes and the presence of cells similar to fibrocytes in the most damaged areas (ANOVA; disease factor: F(1,27) = 1.713; *p* = 0.2016, treatment factor: F(4,27) = 9.142; *p* = 0.0001 and interaction: F(4,27) = 2.653; *p* = 0.0548). The increase in chondrocyte hypertrophy was even more severe in the control group treated with Np-Q (see Figure 9). The results of the experimental rats treated with Np-GSH showed preserved chondrocyte morphology in the same way as with HA.

### 3.9. The Tide Mark

The “tide mark”, or basal line, is the border that delimits the area of articular cartilage from calcified cartilage before coming in contact with the subchondral bone. The tide mark’s loss indicates that the articular damage has reached the deepest strata of the cartilage. It is the beginning of the changes in the subchondral bone that will later give rise to ossification.

The administration of Np-Q was shown to induce a severe lesion in both the control and ACLT groups (Figure 8). Interestingly, this effect was not observed in the groups exposed to Np-GSH and their control group. However, the baseline was slightly interrupted, which was not observed in the other control groups. In the groups with ACLT, it was those treated with HA which showed the most severe damage to the baseline, while the NP- GSH demonstrated better protection of this layer of cartilage, even compared with the control (see Figure 9) (ANOVA; disease factor: F(1,29) = 19.52; *p* = 0.0001, treatment factor: F(4,29) = 4.667; *p* = 0.005 and interaction: F(4,29) = 5.479; *p* = 0.0021).

### 3.10. Subchondral Bone

Like the changes observed in the baseline, differences were observed at the subchondral bone level related to the propagation of mechanical stress in the deeper layers (Figure 8 and Figure 9).

Associating what was found in the radiographic evaluation with the basal status, the Np-GSH did not show any ability to protect this layer, and we even observed changes in the subchondral bone of the control group, contrary to the findings in articular cartilage (ANOVA; disease factor: F(1,27) =14.51; *p* = 0.007, treatment factor: F(4,27) =8.424; *p* = 0.0001 and interaction: F(4,27) = 2.923; *p* = 0.038). 

### 3.11. Articular Surface

In the case of acute lesions, the changes observed on the articular surface of the ACLT groups were minimal, with slight irregularities on the surface almost indistinguishable from the control groups (Figure 8) (ANOVA; disease factor: F(1,29) =9.594; *p* = 0.0043, treatment factor: F(4,29) =84.61; *p* < 0.0001 and interaction: F(4, 29) =5.125; *p* = 0.003). Only the Np-Q showed more severe lesions, both in the control and ACLT groups, even to the point of the observation of severe ulcers in the articular cartilage, as shown in Figure 10.

### 3.12. Extracellular Matrix (MEC)

In the case of the constitution of the cartilage matrix, the changes between the control and ACLT groups were more evident than those on the surface (see Figure 10) (ANOVA; disease factor: F(1,29) = 9.594; *p* = 0.0043, treatment factor: F(4,29) = 84.61; *p* < 0.0001 and interaction: F(4,29) = 5.125; *p* = 0.003). Again, the Np-Qs stand out, showing a greater ECM change towards cartilage with fibrous tissue characteristics. Contrary to these results, Np-GSH showed better matrix preservation.

### 3.13. Acridine Orange

Additionally, staining with acridine orange was performed, as shown in Figure 11, which allowed us to evaluate early cell damage associated with processes of necrosis and apoptosis in the cell population of cartilage [27]; the results demonstrated a decrease in the red coloration in the cytoplasm and nucleus coinciding with an increase in cell death; this is due to the reduction in the concentration of RNA in the cell with DNA degradation.

In the groups with ACLT, less coloration was observed, along with cells with completely green nuclei, especially in the areas close to the acellular areas, possibly related to cell damage in that area. Interestingly, the groups treated with Np-GSH were the only ones that showed differences compared to the other groups, which manifested in a better delimitation of the cells based on their coloration.

### 3.14. Loss of Proteoglycans

It has been reported that toluidine blue staining and the Safranin O staining allow us to see the presence of proteoglycans (PGs) within the articular cartilage. In contrast, the intensity of the Safranin O staining is directly proportional to the concentration of proteoglycans, and toluidine blue staining has been related to a better affinity for osteoarthritis cartilage.

The control groups presented a greater coloration, as shown in Figure 10 and Figure 12, where they have the lowest qualification (less loss of proteoglycans). In contrast, the ACLT groups show a loss of coloration, especially in the areas with cellularity loss, which is to be expected since these cells synthesize PGs.

However, the NP groups showed a slight loss, still perceptible in the areas with viable cells of the control group, due to a decrease in intensity, especially in toluidine blue staining. Although only the ACLT group with Np-Q showed significant differences compared to the controls of the other treatments, the control group presented a loss of color comparable to that of the ACLT groups with Hartmann solution, acetic acid, and HA, while the loss of Np-GSH coloration was more moderated (ANOVA; disease factor: F(1,27) = 48.46; *p* < 0.0001, treatment factor: F(4,27) = 7.583; *p* = 0.0003 and interaction: F(4,27) = 1.316; *p* = 0.2892).

### 3.15. Collagen 2 and Metalloproteinase 13

Col-2 is related to the presence of healthy tissue, which hypertrophies after damage and begins to be replaced by type X collagen. On the other hand, the presence of metalloproteinase 13 (MMP-13) is the most used marker of inflammation in the evaluation of cartilage, as it is the main protease in articular cartilage degradation. The presence of Col-2 was estimated by immunostaining (Figure 13). The results showed a heterogeneous distribution of collagen 2 between individuals, and no significant differences were found between groups (ANOVA; disease factor: F (1,29) =0.4921; *p* = 0.4886, treatment factor: F(4,29) = 1.132; *p* = 0.3608 and interaction: F (4,29) =0.3968; *p* = 0.8092).

Some observed characteristics were not considered in the statistical evaluation, such as the organization of the fibers. We found the most remarkable difference between groups with ACLT; the fibers showed an altered organization compared with their controls.

Additionally, the treatments with Np-Q showed an increase in the intensity of immunoreactivity. At the same time, they seemed to exhibit an agglomeration of the collagen fibers, while the Np-GSH conserved a better organization of the fibers despite showing the most intense immunoreactivity and best organization of fibers, even with respect to the control group of Hartmann solution not increasing the signal. Interestingly, the HA, control, and ACLT groups showed the most intense signal and the best organization of the fibers, even compared to the control group of Hartmann solution. 

Metalloproteinase 13 (MMP-13) is one of the inflammation markers in OA since it is one of the leading causes of cartilage degradation. MMP-13 is detectable in healthy joints since it participates in the normal turnover of the ECM [19,21]. However, it has been reported to increase drastically from the first week after the induction of OA by ACLT [24].

The control groups with Hartmann solution, acetic acid, and HA showed no signs or one or two isolated points in the joint (Figure 13), which does seem to be abnormal [28]. Despite this, the NP group had a lower concentration of MMP-13 compared to the HA ACLT group, which had the highest concentration of MMP-13 in the entire study, which confirms that although HA has a mechanoprotective function, it is not so anti-inflammatory (Figure 9) (ANOVA; disease factor: F(1,23) = 11.48; *p* = 0.0025, treatment factor: F(4,23) = 3.669; *p* = 0.0188 and interaction: F(4,23) = 4.203; *p* = 0.0107).

## 4. Conclusions

In osteoarthritis, oxidative damage contributes to chronic inflammation and promotes the secretory phenotype associated with senescence, which is characteristic of an osteoarthritic chondrocyte in terms of cytokines, chemokines, and proteases. Oxidative stress plays an important role in the development of the disease since, in an avascular environment, with low oxygen concentrations, some metabolic functions are limited.

GSH is involved in various cellular functions, such as cell proliferation, apoptosis, and maintenance in the reduced state of the thiol groups of proteins, allowing the generation of various intracellular signaling cascades. But one of the most important functions is its antioxidant capacity since it can act directly on free radicals (ROS) or participate as an enzymatic cofactor. Two mechanisms by which GSH can be transported have been identified. However, there is no evidence that either of these two families of transporter proteins can reincorporate the GSH released in the extracellular space into the cell, so GSH is eventually depleted if not synthesized. 

Considering this cellular limitation and that the cellular events contributing to osteoarthritis involve the oxidation–reduction imbalance and include inflammation, disorganization, and function of proteoglycans and collagen in the ECM, we studied the effect of nanostructured systems with glutathione in rats with surgical osteoarthritis.

In general, except for the group treated with Np-Q, the groups without OA presented some alterations in cartilage morphology, which could be associated with mechanical stress caused by ambulatory compensation of the contralateral knee or inflammation caused by the arthrocentesis procedure.

Acetic acid, which was used as a control for the nanoparticle vehicle, was the only treatment that stimulated an increase in GSH and MDA, suggesting a possible stimulation caused by the pH of the vehicle (pH 4.1). These effects were not significantly associated with cell damage that may have occurred since the histopathological changes observed in acetic acid (the medium in which the nanoparticles under study were dispersed) were similar compared to those presented in Hartmann’s solution.

Zheng-Shun We et al. (2013) worked with Np-Q in acetic acid at a pH of 5 without finding negative effects due to the pH of the solution. Although we have shown that the vehicle can be used at pH 4.1, it is appropriate to consider working with vehicles at physiological pH to avoid possible negative effects observed in biochemical tests [29].

Chitosan has been shown to decrease MDA concentrations while increasing the activity of antioxidant enzymes. These effects are raised to a greater degree when chitosan is nanoparticulated, which would be expected to control the degenerative process of OA in vivo. However, our study observed more remarkable degenerative changes when using only chitosan NPs, even under control conditions.

On the other hand, severe inflammatory reactions to high doses of chitosan and its NPs have been described, suggesting that the harmful effects observed in this study could be due to a high concentration of chitosan above tolerable levels for the organism, inducing a greater inflammatory process associated with the increase in MMP-13, which triggered the joint degradation observed in histopathological studies [30,31,32].

Chitosan-based grafts used in the treatment of OA for its regeneration have proven to be an ideal matrix for chondrocyte proliferation and have similarities with PG and HA in structure and mechanical properties. However, the new tissue tends to be fibrocartilage with a lower concentration of TP and is mechanically less effective than the native tissue [33,34]. Similar effects were observed in this work, where there was more significant fibrosis of the joint tissue with a lower amount of TP.

Even though chitosan shares structural similarities with the glucosamine glycans (GAGs) present in PGs, which have been shown to stimulate chondrogenesis [34], chitosan seems to interact with growth factors to stimulate fibrocytes and no other cells [35,36]. This may explain why despite an increase in the viability of the cells, they had a morphology like that of fibrocytes [22].

Two other effects observed when using intra-articular Np-Q were increased production of Col-2 and a more significant amount of MMP-13. Although Pickarski demonstrated an increase in the production of Col-2 and MMP-13 from week 1 using the ACLT technique, in our case, the presence of these two compounds was significantly higher under the Np-Q stimulus compared to joints with ACLT treated with Hartmann solution and acetic acid [28].

The characteristics of Col 2 observed in the rats with osteoarthrosis and treated with our NPs are the result of the stimulation of a stable synthesis, coupled with the strong interaction of chitosan with collagen, forming a molecular complex that could confer mechanical stability and, in turn, greater protective stability against collagen degradation [30,33,34].

On the other hand, the Np-GSH showed the same positive effects as the Np-Q on the increase in the synthesis of Col-2. Degenerative changes were almost not observed in joints treated with Np-GSH, suggesting a beneficial effect of GSH towards disease regulation and the side effects of chitosan. Overall, Np-GSH managed to protect chondrocytes from apoptosis, which was revealed by a lower cell population loss in the groups treated with these NPs, as well as a better preservation of cell morphology, compared with the NP-Q and a better conservation of the ECM, indicating the capacity of GSH to help these cells to maintain the correct constitution of the ECM.

The most evident changes induced by Np-GSH were those within the subchondral bone, both in radiographs and histopathological sections, and these changes were similar to the changes observed with Np-Q. The deeper layers of the joint, compared to the articular surface, lead us to think about the possible inability of Np-GSH to reach these layers, remaining trapped exclusively in the articular cartilage, possibly due to the positive charge of the NPs that interact with the negatively charged ECM, so its therapeutic effect is limited to the ECM in its superficial layers and cannot reach the subchondral bone. The question of the behavior of Np-GSH on the subchondral bone in cases of advanced OA arises here, where the erosion of the cartilage exposes this structure [22,37].

As in other studies, HA demonstrated the ability to regulate the OA process without any anti-inflammatory effect. The changes observed were not significantly different concerning animals treated with Hartmann solution. However, the most important result of HA is reported in the long term, with 2 weeks being insufficient to see a difference concerning untreated joints with OA [38,39,40].

It is worth mentioning that in the case of nanoparticle exposure, the effect evaluated is subacute. To thoroughly compare the effectiveness of the treatments proposed in this work, we consider that it is necessary to study the cellular organization of cartilage and the cellular signals associated with the modulation of oxidative stress by nanovectorized GSH, given the beneficial changes observed in situ after administering a single dose of Np-GSH, suggesting the ability to exert biological effects associated with cell proliferation, remodeling of the extracellular matrix, and modulation associated with GSH-dependent antioxidation.

## Figures and Tables

**Figure 1 pharmaceutics-15-02172-f001:**
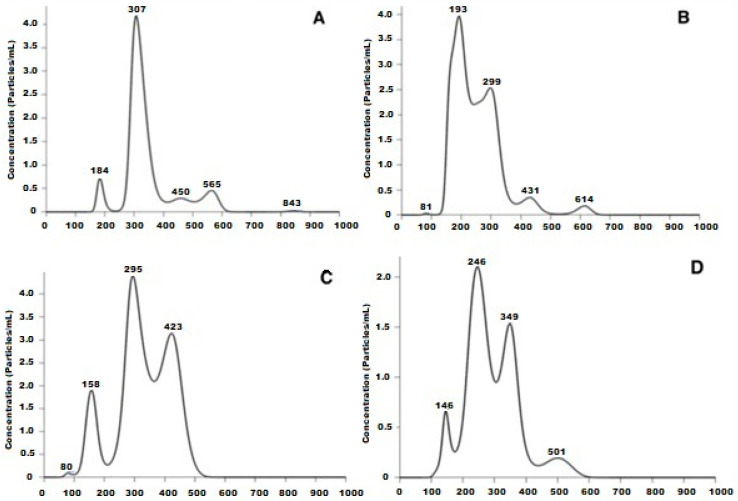
Size distribution of the different nanoparticles (the scale on the graph is ×10^−10^): (**A**) Np-GSH before sterilization. (**B**) Np-GSH after being exposed to UV light for 14 h. (**C**) Np-Q before sterilization with UV light. (**D**) Np-Q after being exposed to ultraviolet light for 14 h.

**Figure 2 pharmaceutics-15-02172-f002:**
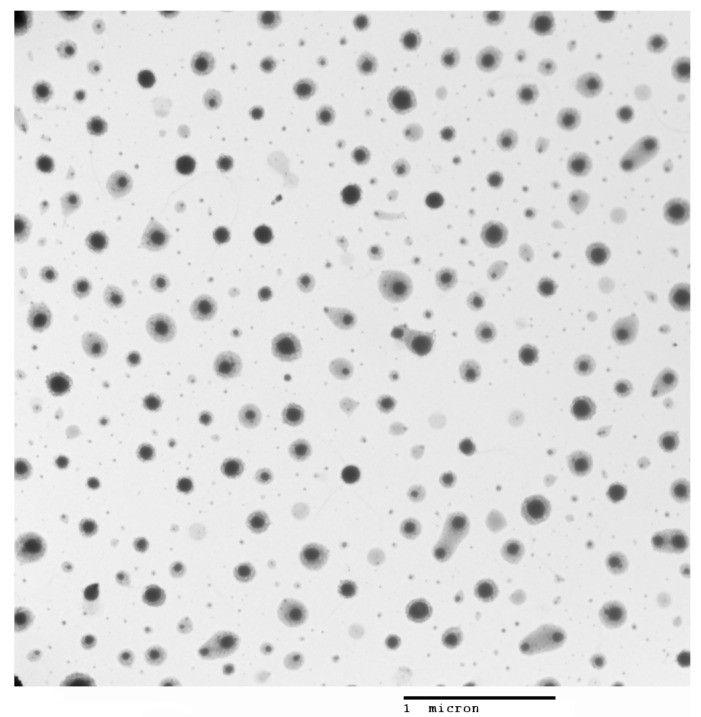
Transmission electron microscopy of chitosan–glutathione nanoparticles.

**Figure 3 pharmaceutics-15-02172-f003:**
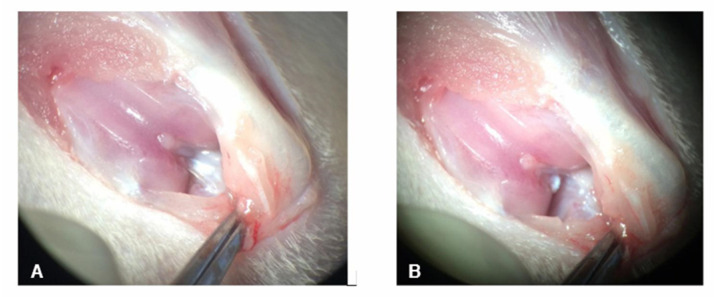
Photograph of the knee during ACLT. (**A**) The cranial cruciate ligament is observed in the background, still intact, and (**B**) after its transection.

**Figure 4 pharmaceutics-15-02172-f004:**
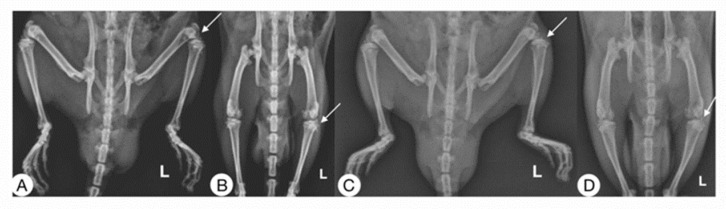
Radiographic images (mediolateral and anteroposterior) of rats 14 days post-surgery are shown. Right pelvic limb control and left ACLT. (**A**,**B**) Rat treated with Hartmann solution. (**C**,**D**) Rat treated with Np-GSH. The arrows show where the transection was carried out in each case.

**Figure 5 pharmaceutics-15-02172-f005:**
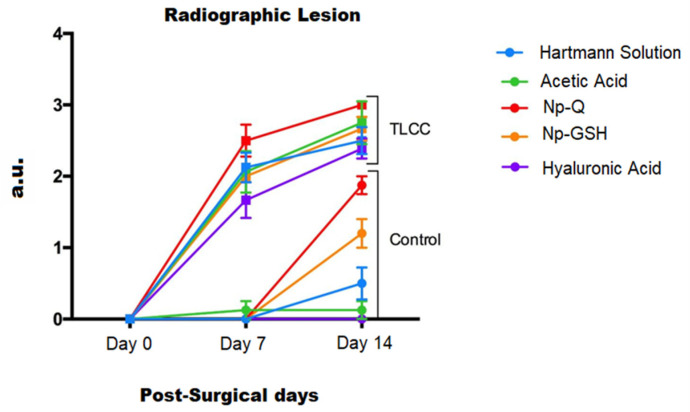
Analysis of injury induced by different treatments according to the Kellgren and Lawrence scale (ANOVA; F(9,80) 79.19; *p* ≤ 0.001).

**Figure 6 pharmaceutics-15-02172-f006:**
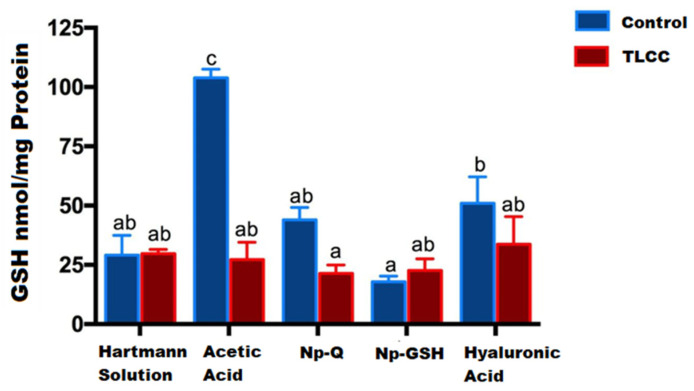
GSH concentration in cartilage and menisci of rats at 14 days post-surgery and nanoparticles exposed exposure. The mean ± SEM of the GSH concentration is shown. Bars with the same letter are not significantly different (ANOVA; F(9,24) = 15.11; *p* < 0.0001).

**Figure 7 pharmaceutics-15-02172-f007:**
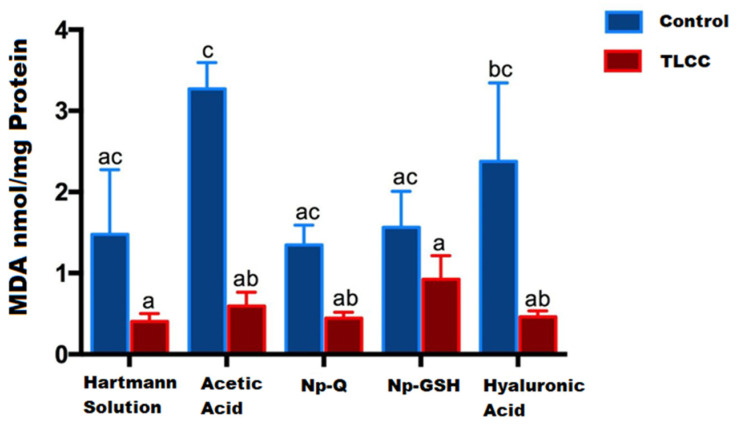
MDA concentration in cartilage and menisci of rats 14 days post-surgery and nanoparticles exposed exposure. The mean ± SEM of the concentration of MDA is shown. Bars with the same letter are not significantly different (ANOVA; F(9,26) = 5.623; *p* < 0.0003).

**Figure 8 pharmaceutics-15-02172-f008:**
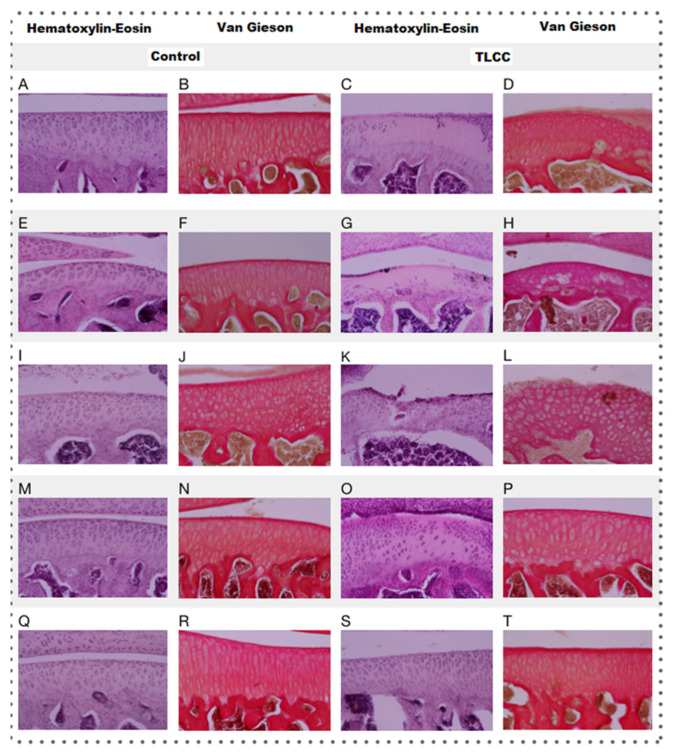
Histopathology analysis of the treated rat knees (20×) using HE and van Gieson staining. The shown letter represents each treatment administered to experimental animals as follows: Hartmann solution (**A**–**D**), acetic acid (**E**–**H**), Np-Q (**I**–**L**), Np-GSH (**M**–**P**), and hyaluronic acid (**Q**–**T**).

**Figure 9 pharmaceutics-15-02172-f009:**
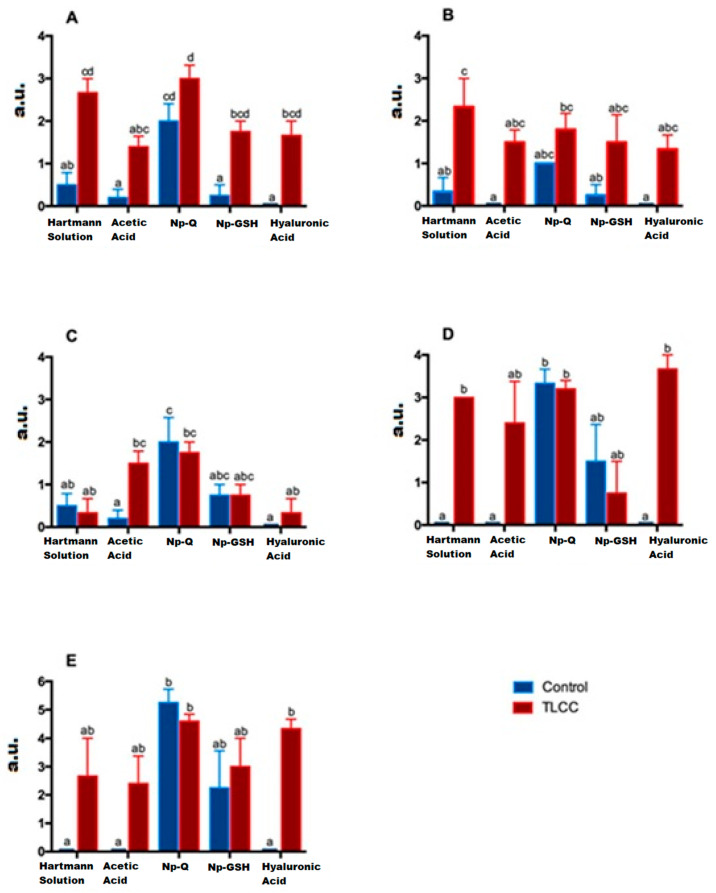
Effect of the different treatments on morphological characteristics evaluated using a modified Mankin score in arbitrary units (a. u.) (Cellular cartilage distribution (**A**), Cellular morphology (**B**), Cellular population (**C**), Tide mark (**D**), and Subchondral bone (**E**)). Bars with the same letter are not significantly different (ANOVA; treatment factor: F(4,30) = 13.52; *p* < 0.0001 and interaction: F(4,30) = 1.191; *p* = 0.3348).

**Figure 10 pharmaceutics-15-02172-f010:**
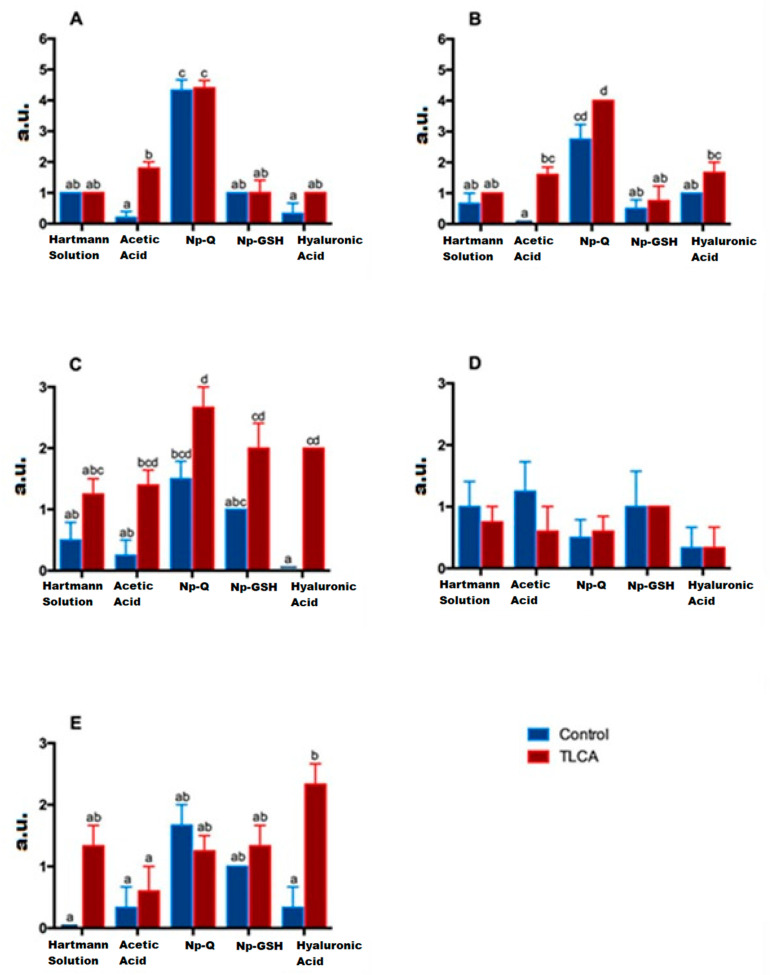
Effect of the different treatments on morphological characteristics of the cartilage (surface (**A**), MEC (**B**), loss of PG (**C**), Col-2 (**D**), and MMP-13 (**E**)). The analysis was done by indirect immunostaining using arbitrary units (a. u.) according to the observed signal as follows, 0: no signal, 1: weak signal, 2: moderate signal, 3: marked signal, Greater than 3: greater signal than marked. Bars with the same letter are not significantly different (ANOVA; disease factor: F(1,29) = 9.594; *p* = 0.0043, treatment factor: F(4,29) = 84.61; *p* < 0.0001 and interaction: F(4,29) = 5.125; *p* = 0.003).

**Figure 11 pharmaceutics-15-02172-f011:**
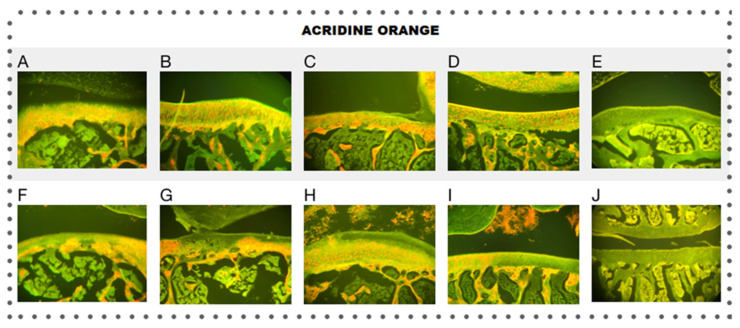
Histology of the knees (10×), acridine orange staining of the different treatments for control (**A**–**E**) and ACLT (**F**–**J**): Hartmann solution (**A**,**F**), acetic acid (**B**,**G**), Np-Q (**C**,**H**), Np-GSH (**D**,**I**), and hyaluronic acid (**E**,**J**).

**Figure 12 pharmaceutics-15-02172-f012:**
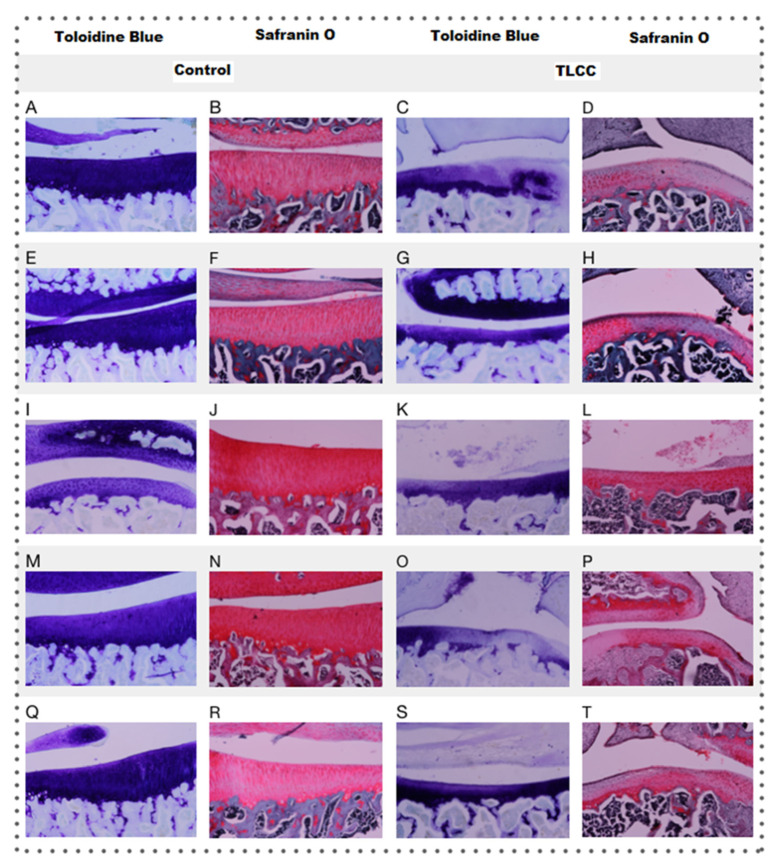
Histopathology of the knees (10×), toluidine blue and Safranin O staining of the different treatments: Hartmann solution (**A**–**D**), acetic acid (**E**–**H**), Np-Q (**I**–**L**), Np-GSH (**M**–**P**) and hyaluronic acid (**Q**–**T**).

**Figure 13 pharmaceutics-15-02172-f013:**
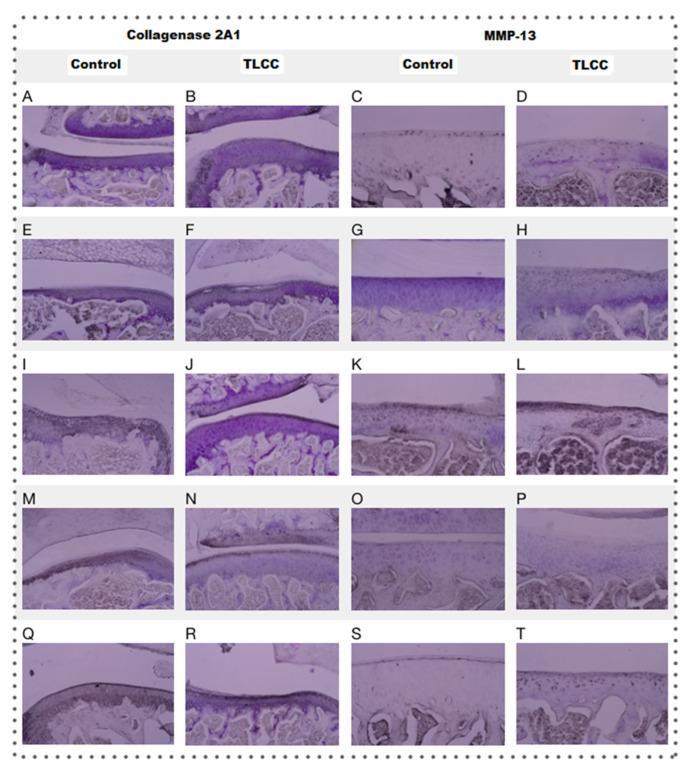
Micrograph of the knees immunostained for collagen 2A1 (10×) and metalloproteinase-13 (20×) for the different treatments: Hartmann solution (**A**–**D**), acetic acid (**E**–**H**), Np-Q (**I**–**L**), Np-GSH (**M**–**P**) and hyaluronic acid (**Q**–**T**).

**Table 1 pharmaceutics-15-02172-t001:** Treatments of each experimental group.

Group	Treatments
1	Hartmann Solution (Pisa^®^)
2	Acetic Acid 1% pH 4.1 (Nanoparticle Medium)
3	Chitosan Nanoparticles (Np-Q)
4	Chitosan/Glutathione Nanoparticles (Np-GSH)
5	Hyaluronic Acid (Legend^®^)

**Table 2 pharmaceutics-15-02172-t002:** The nanoparticles’ mean size, PDI, and zeta potential before and after their sterilization with UV light. The concentration of NPs/mL in the different phases is shown.

	Chitosan–Glutathione Nanoparticles		Chitosan Nanoparticles	
	Pre-UV	Post-UV	Pre-UV	Post-UV
Mean (nm)	354.3	253.3	330.0	292.1
PDI	0.306	0.444	0.313	0.346
Nanoparticles/mL	3.34 × 10^10^	5.51 × 10^10^	1.49 × 10^10^	3.15 × 10^10^
Zeta Potential (mV)		24.8		32.6

## Data Availability

Data supporting the reported results of this paper can be found in the Multidisciplinary Research Unit, Facultad de Estudios Superiores Cuautitlán, Universidad Nacional Autónoma de México, Carretera Cuautitlán-Teoloyucan Km. 2.5, San Sebastián Xhala, Cuautitlán Izcalli, Estado de México, México, CP 54714 1, in Iliane Zetina’s work log.

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
