# Peer review of "Study of the Early Effects of Chitosan Nanoparticles with Glutathione in Rats with Osteoarthrosis"

_pharmaceutics, 2023, doi:10.3390/pharmaceutics15082172_

Round 1
Reviewer 1 Report
Osteoarthritis is characterized by progressive erosion of the articular cartilage along with inflammatory factors. Using antioxidant compounds may be a practical way to treat osteoarthritis.
Glutathione (GSH) is the primary cellular antioxidant. The manuscript administration of chitosan nanoparticles with or without glutathione in an animal osteoarthritis model (TLCC).
Some useful information includes more remarkable degenerative changes when using only Chitosan NP, especially high concentrations of Chitosan.
However, most of the experiments have no or slightly observable difference between the control and the GSH-NP groups.
It is hard to conclude the benefit of GSH-NP for the treatment of osteoarthritis from the current data.
English is fine
Reviewer 2 Report
The manuscript of P. Ramizer-Noguera et al. entitled "Study of the early effects of chitosan nanoparticles with glutathione in rats with” aims to evaluate the nanoparticles capacity in rats with induced surgical osteoarthritis. It is a well written manuscript, the application of glutathione-based chitosan nanoparticles is novel for osteoarthrosis treatment, but there are already several publications based of chitosan nanoparticles dedicated to assess the chondroprotective influence of chitosan in an induced experimental osteoarthritis (OA) model (doi: 10.3390/molecules25235738) and antioxidant activity of glutathione loaded into chitosan nanoparticles for topical administration. Although some interesting data have been presented, several issues need to be addressed and added before this manuscript could be considered for publication. I would recommend a resubmission with major revision based on the following general comments:
In general terms, it is not understood why the study is carried out with nanoparticles of chitosan (Np-Q) and chitosan with glutathione (Np-GSH) suspended in a 1% acetic acid solution, in addition to being compared with a 1% acetic acid control. In all three cases, the effect of acetic acid will mask the results of the study due to its high oxidizing and solubilizing power. Likewise, chitosan at pH 4.1 will be protonated and the nanoparticles are probably highly degraded since chitosan degrades very quickly at 37ºC and acidic media. The treatment of the nanoparticles is not described, but the most optimal would be to prepare them, clean the reagents and suspend them in a physiological medium for their evaluation.
Line 72. Although the synthesis of the NP is described in another article, it would be convenient to describe them briefly. At least one should write what the NPs are made of, for example what the chitosan has been cross-linked with and how GSH has been encapsulated in the nanoparticle. Likewise, a morphological characterization of the nanoparticles by electronic microscopy would need to be carried out.
Table 1. I don't know if it makes sense to make a blank with a 1% acetic solution since the medium of the nanoparticles probably does not contain 1% acetic acid after the formation of the nanoparticles. 1% acetic would probably dissolve the NPs formed.
Table 2. It would be good to put the polydispersity index (PdI) of the samples and the zeta potential in the table. Chitosan nanoparticles have a positive charge; therefore, it would be interesting to know the charge of the nanoparticle once the glutathione is incorporated.
Figure 1. The text on the ordinate axis cannot be read correctly. Based on the particle size distribution profiles, the samples appear very heterogeneous, as up to four different populations of particles are observed. Do the authors believe that the different sizes of nanoparticles can affect their therapeutic effect?
Figures 8 and 9. The text cannot be read correctly.
Reviewer 3 Report
Review on the manuscript entitled “Study of the early effects of Chitosan Nanoparticles with Glutathione in rats with Osteoarthrosis” By Patricia Ramírez-Noguera et al.
The Manuscript describes the capability of Chitosan Nanoparticles with Glutathione (Np-GSH) to regulate the oxide-redox status in in vivo using Wistar rats with induced surgical osteoarthritis. Radiographic, biochemical (GSH and TBARS 20 quantification), histopathological, and immunohistochemical (Col-2 and MMP-13) analyses were evaluated to see the progress of the osteoarthritic lesions after the administration of a single dose of Np-GSH. According to the results obtained, the GSH contained in the NP could be vectored to chondrocytes and used by the cell to modulate the oxidative state reduction, decreasing the production of ROS and free radicals induced by agents oxidizing xenobiotics and decreasing lipid peroxidation. The results allow to consider the nanostructures developed as a helpful to reduce the damage associated with oxidative stress in osteoarthritis.
General comments of the reviewer:
1. There is a lack of a brief discussion of tactics at the beginning of each section (the purpose and idea of the experiment) and a discussion – report on the results at the end of each section. Otherwise, it is difficult to follow the logic of the study.
2. There are not logical transitions between sections.
3. Synthesis of Np-GSH particles – although it is published by the authors earlier – it is necessary to discuss the synthesis scheme in the Methods section and in the results section – characteristics, properties and molecular structure of the resulting particles should be discussed (whether glutathione is covalently included, which method of gelation, which chitosan was used, what MM, and this is not indicated in the original article- ref 13, etc.)
4. In Conclusions section: It is necessary to structure the text and highlight the role of glutathione more clearly : in decreasing the production of ROS and free radicals induced by agents oxidizing xenobiotics
5. In Conclusions section: it is better not to use abbreviations so that they are independent of the rest of the text
6. It makes sense to introduce the abbreviations section or ease of perception
Extensive editing of English language required
Round 2
Reviewer 1 Report
Even the histochemistry staining showed some positive results. The molecular change precedes the morphological changes, but the other data showed little or no benefit. It is difficult to conclude the effect of treatment.
Good writing in English
Reviewer 2 Report
The authors have made the pertinent changes to improve the quality of the manuscript. The materials and methods section has been improved, and the figures as well. I still don't understand why they work at pH 4 and why they use an acetic acid solution as a control. In general, the manuscript has been considerably improved and could be acccepted in the present form.
Reviewer 3 Report
most of the recommendations were taken into account by the authors
Typo and grammatical errors should be revised.